# Camelpox Virus in Western Kazakhstan: Isolation and Phylogenetic Analysis of a New Strain

**DOI:** 10.3390/v17091229

**Published:** 2025-09-09

**Authors:** Yerbol Bulatov, Nurlan Kozhabergenov, Zhanat Amanova, Zhanna Sametova, Zhanat Kondybaeva, Ruslan Abitayev, Abdurakhman Ussembay, Alina Kurmasheva, Dariya Toktyrova, Dana Mazbayeva, Kuandyk Zhugunissov, Muratbay Mambetaliyev, Bekbolat Usserbayev, Sholpan Turyskeldy

**Affiliations:** 1Research Institute for Biological Safety Problems, National Holding QazBioPharm, Gvardeiskiy 080409, Kazakhstan; ye.bulatov@biosafety.kz (Y.B.); n.kozhabergenov@biosafety.kz (N.K.); zh.amanova@biosafety.kz (Z.A.); zh.sametova@biosafety.kz (Z.S.); zh.kondybaeva@biosafety.kz (Z.K.); r.abitaev@biosafety.kz (R.A.); a.ussenbay@biosafety.kz (A.U.); a.kurmasheva@biosafety.kz (A.K.); d.toktirova@biosafety.kz (D.T.); mazbayevad@mail.ru (D.M.);; 2Al-Farabi Kazakh National University, Almaty 050040, Kazakhstan

**Keywords:** Camelpox virus, *Orthopoxvirus*, isolate, phylogenetic analysis, reservoirs

## Abstract

This study continues earlier work aimed at identifying potential natural reservoirs of camelpox virus (CMLV) during interepizootic periods. In 2023–2024, field expeditions in western Kazakhstan led to the collection and analysis of biological samples from camels, rodents and hematophagous insects. Despite the absence of clinical symptoms, PCR-positive results were obtained from camel blood samples. These samples underwent molecular genetic analysis, including viral DNA detection and whole-genome sequencing. Using next-generation sequencing, the complete genome of the *Camelpox virus/Beineu/2023* isolate (202.273 bp) was obtained and deposited in the NCBI database (accession number PV920573.1). The isolate showed >98% genetic similarity to the previously described Kazakhstan strain *M-96*, indicating long-term local circulation of a genetically stable variant. Phylogenetic analysis confirmed the isolate’s evolutionary conservatism and close relationship with other CMLV strains. The findings suggest that camels serve as a natural reservoir, enabling viral persistence and potential reactivation under stress-related conditions. The observed geographic clustering underscores the need for region-specific molecular surveillance to ensure timely detection of new variants and prevent cross-border spread.

## 1. Introduction

Camelpox is an infectious viral disease that is widespread among camel populations. It is endemic in the Middle East (Iran, Iraq, Saudi Arabia, the United Arab Emirates, Yemen), Asia (India, Afghanistan, Pakistan), Africa (Algeria, Egypt, Kenya, Mauritania, Niger, Somalia, Morocco, Ethiopia, Oman, Sudan), and in the southern regions of the former USSR. The causative agent is the camelpox virus (CMLV), a member of the genus *Orthopoxvirus* within the family *Poxviridae* [1].

Camelpox is characterized by fever and localized or generalized lesions of the skin and mucous membranes, including those of the oral cavity, as well as the respiratory and digestive tracts. Transmission occurs through direct contact with infected animals, either via skin abrasions or aerosolized particles. Infectious materials such as scabs, saliva, and secretion from infected camels can contaminate the environment, including water sources, which may act as temporary reservoirs of infection. Severe outbreaks in young and previously unexposed camels are often associated with high mortality rates [2,3]. Studies indicate that the frequency and severity of outbreaks increase during the rainy season, whereas milder cases are predominantly observed during the dry season. Both morbidity and mortality rates are higher in male camels than in females. In adult camels, mortality ranges from 10% to 28%, while in young animals it may reach 25–100%. Variations in the severity of clinical manifestations are thought to be related to differences among circulating CMLV strains [2,3,4]. Additionally, camelpox has been described as a potential zoonosis, with three confirmed cases of human infection reported in India [5,6,7].

Phylogenetic analysis has shown that among viruses of the genus *Orthopoxvirus*, the CMLV is most closely related to the variola virus, the causative agent of smallpox, although both viruses infect a strictly limited range of hosts. Comparative analysis of the nucleotide sequences of the CMLV genome confirmed this close relationship, revealing a nucleotide identity of 96.6–98.6% between CMLV and Variola virus. Furthermore, a DNA distance matrix indicated a lower genetic distance between CMLV and Variola virus compared to that between CMLV and the vaccinia virus [8,9,10,11,12]. The global and regional prevalence of camelpox, including in Kazakhstan, has been assessed using data from the World Organization for Animal Health (OIE). Between 2017 and 2021, ten outbreaks of camelpox were reported worldwide, occurring in India, Iran, Iraq, Afghanistan, Pakistan, Saudi Arabia, several African countries, Russia. Turkmenistan and Kazakhstan [13]. According to veterinary records from the Republic of Kazakhstan, sporadic cases of camelpox were recorded in the Mangistau and Atyrau regions in 1930, 1942–1943, 1965–1969, 1996 and 2019–2020. Statistical analysis of these data suggests a cyclical pattern of disease occurrence, with an interval of approximately 10 to 20 years [14,15].

Given the above, the question of camelpox virus localization during inter-epidemic periods remains highly relevant. Identifying its natural reservoir is crucial for preventing potential outbreaks and enabling the timely implementation of preventive measures.

This work represents a continuation of previously initiated study aimed at identifying the natural reservoirs of the camelpox virus. The study involved the analysis of samples collected from camels, rodents and hematophagous insects in Western Kazakhstan. Despite the absence of pronounced clinical signs, camels tested positive during the investigation. According to the literature, camelpox may occur in latent or subclinical forms, particularly among adult animals in endemic regions [16,17].

The collected samples were subjected to further molecular genetic analysis, including detection of viral DNA and whole-genome sequencing. Based on the obtained nucleotide sequences, phylogenetic analysis was carried out to determine the genetic relationships between the isolated strains and other known CMLV strains, and to propose potential routes of virus circulation in natural foci.

## 2. Materials and Methods

### 2.1. Inoculation of Positive Samples into the Chorioallantoic Membrane of Chicken Embryos

Whole blood samples were initially centrifuged at 3000 rpm for 10 min. The resulting supernatant was used to inoculate 10-day-old specific pathogen-free (SPF) chicken embryos. Inoculation was carried out under sterile conditions by administering 0.2 mL of the sample into the chorioallantoic membrane (CAM). The inoculated embryos were incubated at 37 °C with a relative humidity of 60–70% for 5–7 days. Embryo viability was monitored daily.

Embryos that died within the first 24 h post-inoculation were excluded from further analysis. Following the incubation period, the embryos were chilled at 4 °C for 24 h. Subsequently, the eggs were opened, and the CAMs were examined for pathomorphological changes indicative of viral infection, including the presence of specific pock-like lesions.

### 2.2. Inoculation of the Isolated Virus into Cell Cultures

For the cultivation of camelpox virus, kidney cell cultures from the African green monkey (*Vero*) and lamb kidney cells were used. The *Vero* cell line (Vero C1008; ATCC^®^ CRL-1586™) was obtained from the American Type Culture Collection (ATCC, Manassas, VA, USA) https://www.atcc.org/products/crl-1586 (accessed on 7 July 2025). Cells were maintained at 37 °C in a humidified atmosphere containing 5% CO_2_ in Dulbecco’s Modified Eagle Medium (DMEM) supplemented with 10% fetal bovine serum (FBS) (Capricorn Scientific, Westport, CT, USA).

Primary cell cultures were obtained from the Laboratory of Cell Biotechnology at the Research Institute for Biological Safety Problems (RIBSP, Taraz, Kazakhstan). Cells were maintained at 37 °C in a 5% CO_2_ atmosphere in semi-synthetic mural medium (SNM, RIBSP, RK) supplemented with 10% fetal bovine serum (Capricorn Scientific, Westport, CT, USA). In this study, primary lamb kidney (LK) cell cultures were derived from animals that were housed and handled in compliance with international and national bioethical standards (Rules for Conducting Biomedical Research on Animals in the Republic of Kazakhstan, 442/2010 and EU Directive 2010/63/EU), and all procedures were performed in accordance with a protocol approved by the Bioethics Council of RIBSP (protocol code 1511/011, dated 15 November 2022).

Cell cultures were infected with viral material previously isolated from the chorioallantoic membrane of chicken embryos. The material was centrifuged at 3000 rpm for 10 min, and the resulting supernatant was filtered through a 0.22 μm pore-size membrane filter. Subsequently, 0.2 mL of the virus-containing filtrate was inoculated onto confluent monolayers of lamb kidney and Vero cell lines. The infected cultures were incubated at 37 °C in a humidified atmosphere with 5% CO_2_ in maintenance medium containing 2% FBS and 6 mM L-glutamine (final concentration 1%) for 5–7 days.

Virus adsorption was carried out at 37 °C for 60 min. After absorption, the inoculum was removed, the cell monolayers were washed with phosphate-buffered saline (PBS), and fresh growth medium was added. The infected cultures were incubated at 37 °C in a humidified atmosphere containing 5% CO_2_ for 5–7 days. The cultures were monitored daily for the development of a cytopathic effect. Serial passaging was performed when necessary. Viral identification was conducted using polymerase chain reaction (PCR) with primers specific for camelpox virus.

### 2.3. Spatial Analysis and Mapping

Spatial analysis, map construction and the application of cartographic symbols and design elements were performed using Esri ArcGIS Pro software, version 2.2. The software tools were employed to visualize sampling locations and to spatially interpret the obtained results.

### 2.4. Isolation of Viral DNA

Viral material obtained from infected cell cultures was first clarified by centrifugation at 2000× *g* for 10 min to remove cellular debris. The resulting supernatant was then concentrated by ultracentrifugation at 200,000× *g* for 20 min at 4 °C. A 200 μL aliquot of the concentrated viral suspension was used for DNA extraction using the QIAamp DNA Mini Kit (Qiagen, Hilden, Germany), following the manufacturer’s instructions. The concentration of the extracted DNA was measured using the Qubit^TM^ DNA High Sensitivity (HS) Assay Kit (Life Technologies, Carlsbad, CA, USA) on a Qubit^TM^2.0 Fluorometer (Life Technologies, Carlsbad, CA, USA), in accordance with the manufacturer’s protocol.

### 2.5. Preparation of DNA Libraries for Next-Generation Sequencing (NGS)

For library preparation, 20 μL of viral DNA at a concentration of 100 ng/μL were used. DNA fragmentation was performed using the Ion Shear^TM^ Plus Kit (Life Technologies, Carlsbad, CA, USA), followed by adapter ligation with the Ion Plus Fragment Library Kit (Thermo Fisher Scientific, Waltham, MA, USA). Purification steps were carried out using AgencourtAMPure XP magnetic beads (Beckman Coulter, Brea, CA, USA) according to the manufacturer’s protocol. The resulting DNA fragments were separated by horizontal electrophoresis in a 1.5% agarose gel (Sigma-Aldrich, St. Louis, MO, USA) stained with SYBR^TM^ Safe DNA fragments ranging from 350 to 500 base pairs were excised from the gel, visualized, and documented using the iBright ^TM^ CL1500 Imagine System (Thermo Fisher Scientific, Waltham, MA, USA). Gene Ruler ^TM^ 1 kb Plus DNA Ladder (Thermo Fisher Scientific, Waltham, MA, USA) was used as a molecular weight marker. Gel-purified DNA fragments were extracted using the innuPREPDOUBLEpure Kit (Analytik Jena AG, Jena, Germany).

DNA libraries were amplified using components of the Ion Plus Fragment Library Kit following the manufacturer’s instructions. The concentration of the amplified libraries was quantified using the Ion Universal Library Quantitation Kit (Thermo Fisher Scientific, Waltham, MA, USA). Final preparation of the libraries was carried out on the Ion Chef ^TM^ System using the Ion 510^TM^, and 530^TM^ Kit together with the Ion 530^TM^ Chip (Thermo Fisher Scientific, Waltham, MA, USA).

### 2.6. Next-Generation Sequencing (NGS) and Data Analysis

Whole-genome sequencing of the camelpox virus was performed using the Ion Torrent platform on the Ion GeneStudio™ S5 System (Thermo Fisher Scientific, Waltham, MA, USA). The raw sequencing data were processed using Ion Torrent Suite Software (v5.12) and exported in UBAM and FASTQ file formats. Genome assembly was carried out in the UGENE software environment (v52.1), using the genome of Camelpox virus strain 0408151v (GenBank accession no. KP768318.1) as a reference. FASTQ files served as the input data for downstream analysis. Read quality assessment was conducted using FastQC. Genome assembly was performed with the BWA-MEM algorithm. Annotation of the assembled genome was performed using the Prokka annotation pipeline (Galaxy version 1.14.6+galaxy1), which is primarily designed for prokaryotic genome annotation.

### 2.7. Phylogenetic Analysis

Multiple sequence alignment and phylogenetic analysis were performed using MEGA software version 12 (Mega Cloud Services Lomoted, Auckland, New Zealand) utilizing up to 7 parallel computing threads. The evolutionary history was inferred using the Neighbor-Joining method based on the Tamura-Nei substitution model. The robustness of the phylogenetic tree was evaluated by bootstrap analysis with 500 replicates. The analytical procedure encompassed 10 nucleotide sequences. Only reads with a minimum length of 70 bp and Phred quality score ≥20 were retained for downstream analysis. The pairwise deletion option was applied to all ambiguous positions for each sequence pair.

## 3. Results

### 3.1. Positive Samples

According to previous studies, positive PCR results for the presence of camelpox virus were obtained exclusively from samples collected from camels. No viral genetic material was detected in samples collected from rodents and hematophagous insects. Based on the positive findings, a distribution map of CMLV in the Mangystau and Atyrau regions was generated. Although the concentration of viral material in individual samples may have been low, all PCR-positive cases were included in the spatial analysis due to their potential epizootic significance and the associated risk of virus circulation in these areas. Geographic mapping of CMLV-positive cases enables the identification of high-risk zones, facilitates monitoring of virus circulation dynamics, and supports the development of targeted strategies for epizootic surveillance and biosafety planning.

In the Mangystau region, 21 out of 300 examined camel blood samples tested positive by PCR (Figure 1). The map illustrates the proportion of positive samples by administrative district.

In the Atyrau region, 22 out of 420 tested camel blood samples yielded positive PCR results. The highest proportions of positive samples were recorded in Beineu (32%) and Inder (41%) districts. Lower proportions were observed in Kurmangazy (9%), Makat (4.5%), and Makhambet (9%) districts. No positive samples were detected in the Isatay district (Figure 2). It should be noted that the reported percentage values are calculated relative to the total number of positive samples, rather than the total number of samples tested, due to the relatively low number of PCR-positive cases. This approach allows for a clearer representation of the geographic distribution of virus detections. The aggregated pie chart in the upper left corner of the figure illustrates the overall distribution of positive samples by district.

### 3.2. Virus Isolation

As previously noted, several PCR-positive samples were identified during monitoring. However, successful virus isolation was achieved from only a single sample-blood collected from a young camel during the 2023 expedition in the Beineu district of the Mangystau region. The inability to isolate the virus from the remaining PCR-positive samples is likely attributable to an insufficient concentration of viable virions capable of initiating replication in cell cultures.

It should be noted that the PCR method detects fragments of viral nucleic acids even at low viral titers, including non-infectious, defective, or degraded particles that are incapable of initiating an infectious process. The positive sample was inoculated into the chorioallontoic membrane (CAM) of 11-day-old chicken embryos. Signs of viral replication in the CAM were observed starting from the third passage, manifested as elevated, whitish lesions—either pinpoint or confluent—clearly demarcated from surrounding tissue and featuring characteristic central hemorrhagic inclusions (Figure 3).

Initial virus isolation was performed in the chorioallantoic membrane of chicken embryos, after which the harvested viral material was used to infect susceptible cell cultures to facilitate virus adaptation and replication in vitro.

Following inoculation of the isolated virus into Vero and lamb kidney cell cultures, signs of a cytopathic effect (CPE) began to appear after 72 h of incubation. The CPE was characterized by cell rounding, detachment from the substrate, and disruption of the cell monolayer. Viral DNA was extracted from the infected lamb kidney cell culture at the stage of pronounced CPE, indicating active viral replication in this cell line (Figure 4).

These findings confirm the viability of the isolated virus and its ability to adapt to different cell cultures, thereby providing a foundation for further investigation, including molecular-genetic and antigenic characterization. All experiments were conducted in triplicate to ensure reproducibility and reliability of the results.

### 3.3. Molecular Genetic Analysis

The complete genome of the camelpox virus isolate obtained in this study (*Camelpox virus/Beineu/2023)* was sequenced using next-generation sequencing (NGS) technology. The genomic sequence was deposited in the NCBI GenBank database under the accession number PV920573.1. The total genome length was 202,273 base pairs (bp).

Comparative analysis of the *Camelpox virus/Beineu/2023* genome with those of other members of the *Orthopoxvirus* genus revealed both conserved structural features and species-specific variations, which are important for understanding evolutionary relationships within the group. According to nucleotide sequence analysis performed using the BLASTN tool, the isolate showed 99.95% identity with Camelpox virus strains *0408151v* (KP768318.1) and *CMS* (AY009089.1), as well as 99.76% identity with isolates obtained in the United Arab Emirates in 2021 (MZ300859.1, MZ300856.1, MZ300860.1, MZ300858.1, MZ300857.1) and an isolate from Israel in 2016 (MK910851.1).

Multiple sequence alignment was performed based on the assembled complete genome sequences of camelpox virus isolates. Phylogenetic analysis was carried out using MEGA software version 12, and a phylogenetic tree was constructed using the Neighbor-Joining method (Figure 5). Complete nucleotide sequences from the Genbank database were used for the analysis. The evolutionary distances were computed using the Tamura-Nei method and are in the units of the number of base substitutions per site. The analytical procedure encompassed 10 nucleotide sequences. The pairwise deletion option was applied to all ambiguous positions for each sequence pair, resulting in a final data set comprising 206 183 positions. The tree is drawn to scale, with branch lengths in the same units as those of the evolutionary distances used to infer the phylogenetic tree.

The constructed phylogenetic tree illustrates the evolutionary relationships among ten Camelpox virus isolates, which are primarily grouped according to their geographic origin. The analysis revealed that isolates from the United Arab Emirates and Israel form a distinct phylogenetic clade. In contrast, the *Camelpox virus/Beineu/2023* isolate shows a high degree of genetic similarity to isolates *CMS*, *0408151v*, and the previously described Kazakhstan strain *M-96*.

In addition to the whole-genome comparison, we analysed genes encoding proteins involved in virus–host interactions, including A26L, A33R, B5R, and H3L, which are known to participate in receptor binding and viral entry in *Orthopoxviruses* (Table 1).

Comparative analysis of protein sequences of A26L, A33R, B5R, and H3L of *Camelpox virus/Beineu/2023* (PV920573.1) isolate with other known camelpox virus isolates showed a significant difference in the A26L protein at position 336, where amino acid substitution of Valine to Leucine occurred. Also in this protein A26L at position 374, isolates *0408151v* (KP768318.1), *CMS* (AY009089.1) are identical with the studied isolate, while other isolates lack Aspartic Acid at this position. At position 394 of the same protein, the amino acid Glutamine is present in four isolates (PV920573.1, KP768318.1, AY009089.1, AF438165.1) that are close in phylogenetic analysis, and the amino acid Histidine is present in isolates from the UAE and Israel (MK910851.1, MZ300856.1, MZ300859.1, MZ300860.1, MZ300857.1, MZ300858.1). At position 236 of the A26L protein, the studied isolate differs from isolates KP768318.1 and AY009089.1 in the amino acid Alanine. In the A33R protein, together with the isolates KP768318.1, MK910851.1, there is a deletion at position 1, while in other isolates, the amino acid Methionine is located at this position. Differences are also present in the B5R protein between closely related isolates and isolates from the UAE and Israel, where the amino acids Asparagine and Aspartic Acid differ at position 308. Comparison of the H3L protein also showed differences at positions 6 and 175 with isolates from the UAE and Israel.

## 4. Discussion

Camelpox has a substantial economic impact in regions where camels play a vital role in agriculture. These animals are used not only as draft animals but also as sources of milk, meat, and wool. Although other types of livestock may fulfill similar functions, camels are classified as large ruminants, and the cost of a single camel often equals that of dozens of small ruminants. This makes camels particularly valuable to both farming and nomadic communities. Accordingly, the spread of infections such as camelpox poses a significant threat to livestock production and food security in affected regions. The disease has been reported in nearly all countries where camel husbandry is practiced [18]. An analysis of epizootic data from the past decade indicates that sporadic cases have been recorded in countries such as Israel, Iraq, Eritrea, Kazakhstan, and Turkmenistan. Meanwhile, in eight countries—Iran, Libya, Oman, Palestine, Saudi Arabia, Somalia, Tunisia and Ethiopia—the disease is considered endemic, indicating stable circulation of the virus within camel populations [14,19].

This study presents a molecular genetic analysis of a virus isolate obtained from a clinically healthy camel during monitoring aimed at identifying potential natural reservoirs of the camelpox virus. Although no clinical signs of the disease were observed in animals from the studied regions, camelpox is known to circulate in a latent or subclinical form, particularly among adult animals in endemic areas [16,17,20,21]. Such asymptomatic infection complicates timely detection and highlights the critical importance of molecular diagnostic tools and continuous epidemiological surveillance, even in the absence of overt clinical symptoms. The results of phylogenetic analysis based on whole-genome sequences revealed clear clustering of Camelpox virus strains according to their geographic origin and time of isolation. Isolates obtained in 2020 (D1795/20, D1804/20, D1734/20, D1621/20, D1865/20) formed a distinct clade that is genetically distant from the Kazakhstan isolates. This suggests that these isolates originated from a different epizootic focus, presumably located in Saudi Arabia, and possess characteristic regional genomic features. Their high intragroup identity indicates a recent common ancestor and likely reflects a localized outbreak.

The *Negev2016* isolate, obtained in Israel, occupies a phylogenetically intermediate position between the Middle Eastern and Kazakhstan lineages. This may indicate the existence of potential transmission routes associated with animal migration or regional trade. According to whole-genome sequencing data, the *Negev2016* isolate differs from the Kazakhstan strain *M-96* by 349 single nucleotide polymorphisms (SNPs), with an overall sequence identity of 99.55% [22].

The Camelpox strain *CMS*, isolated in Iran in 1970, is a virulent reference isolate widely used in scientific research. Its genome encodes a unique v-slfn protein (encoded by the 176R gene), which is believed to modulate the host immune response without inhibiting cell proliferation [23]. Phylogenetically, the *CMS* strain is closely related to the Kazakhstan strain *M-96* and genetically distant from vaccine strains. It also shows a high degree of homology with Variola virus, with approximately 98% sequence identity in key genomic regions.

The Camelpox virus strain *0408151v* is a laboratory-adapted reference isolate maintained in the National Collection of Pathogenic Viruses (NCPV, Salisbury, UK) [24]. Its genome comprises 202,289 base pairs and is frequently used in research involving phylogenetic analysis and validation of diagnostic tools. Although the exact geographic origin of this strain is not reported in open-access sources, phylogenetic data suggest that it is closely related to field isolates circulating in the Middle East.

The observed amino acid substitutions in A26L, A33R, B5R, and H3L provide additional molecular evidence supporting the phylogenetic clustering of the *Camelpox virus/Beineu/2023* isolate with historical Kazakhstani strains and its divergence from Middle Eastern isolates. In particular, substitutions in A26L (positions 336, 374, and 394) and in H3L (positions 6 and 175) distinguish Kazakhstani isolates from those circulating in the UAE and Israel, suggesting possible regional adaptation or long-term independent evolution. Proteins A26L, A33R, B5R, and H3L are known to participate in virus–host interactions and play important roles in immunogenicity in orthopoxviruses [25,26], while similar amino acid variations in Vaccinia and Monkeypox viruses have been associated with changes in viral spread, antigenicity, and host range [27,28]. Thus, the unique amino acid features identified in the *Camelpox virus/Beineu/2023* isolate may serve as markers of geographic lineage differentiation and could potentially influence viral biology.

Our study lacks functional investigations, such as virulence or immunogenicity assays, which would provide a more comprehensive understanding of the biological significance of the observed amino acid substitutions. Due to regulatory restrictions and the limited scope of the project, such experiments could not be performed. In addition, although several PCR-positive samples were obtained from camels, successful virus isolation was achieved from only one specimen. This limitation is most likely related to the low concentration of viable virions in the majority of positive samples, as PCR detects fragments of viral nucleic acids, including defective or non-infectious particles. Taken together, these limitations should be considered when interpreting the findings; however, they do not diminish the importance of the molecular and phylogenetic data presented here, which provide valuable insights into the genetic diversity of camelpox virus isolates.

Nevertheless, the results obtained in this study demonstrate that the *Camelpox virus/Beineu/2023* isolate shows a high degree of genetic similarity to the *M-96* strain, which was isolated in Kazakhstan more than 30 years ago. This finding indicates the long-term circulation of an endemic genetic variant of CMLV in the region. The observed sequence identity exceeding 98% highlights the genetic stability and evolutionary conservatism of the virus. These data suggest that the same genetic lineage of CMLV has been maintained locally for decades with minimal evolutionary divergence. Camels are likely to serve as the natural reservoir of the virus during interepizootic periods, supporting its persistence and potential reactivation under stress conditions.

At the same time, the observed geographic clustering emphasizes the importance of region-specific molecular surveillance for the early detection of novel variants and the prevention of cross-border transmission. Furthermore, phylogeographic mapping may help to identify the sources of outbreaks and establish epidemiological links between affected regions.

According to the literature, CMLV does not have a transmissible mechanism and spreads mainly through direct contact with infected animals, affected skin, or care items [4,29,30]. However, it has been noted that during the rainy season, the incidence of the disease increases significantly [3,5]. This probably due to a combination of factors: crowding of animals in shelters, shared use of water sources, maceration of the skin due to high humidity, and a decrease in its barrier function [20,31]. An additional factor may be an increase in the number of insects and ticks, such as *Hyalommadromedarii*, which are capable of mechanically transferring the virus between animals. Also, against the background of seasonal climatic stress and weakened non-specific immunity, the susceptibility of camels to infection increases.

Consequently, the combination of epizootological data, molecular genetic analysis, and literature review suggests that camels may not only be susceptible hosts, but also serve as a natural reservoir of camelpox virus. This is supported by the detection of viral DNA in clinically healthy animals, the limited species specificity of the infection, and the genetic stability of viral isolates. Thus, the persistence of CMLV in camels and its periodic reactivation under unfavorable conditions play a key role in maintaining the long-term circulation of the virus and the cyclic nature of epizootics in endemic regions.

## Figures and Tables

**Figure 1 viruses-17-01229-f001:**
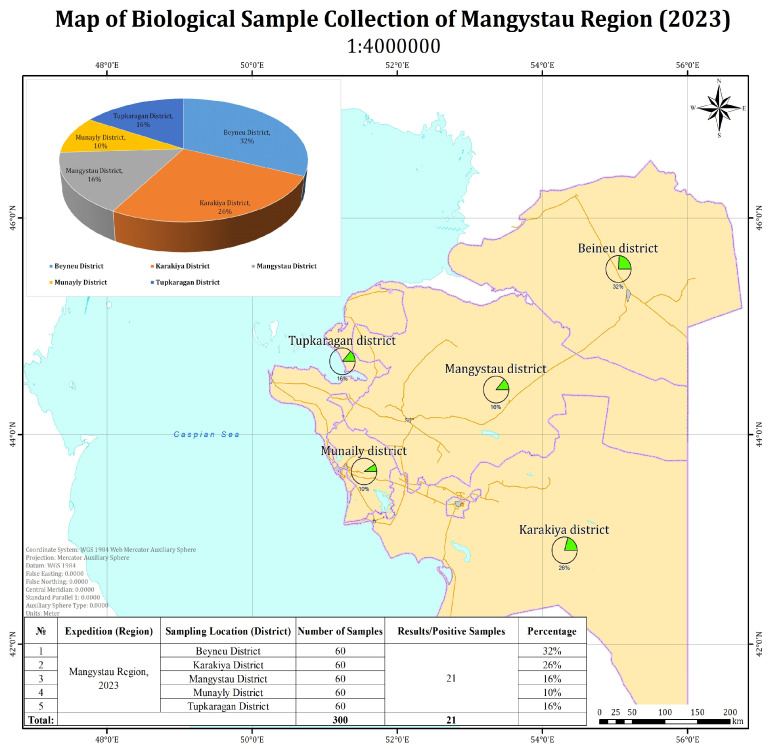
Prevalence of Positive Samples by District in Mangystau Region (2023).

**Figure 2 viruses-17-01229-f002:**
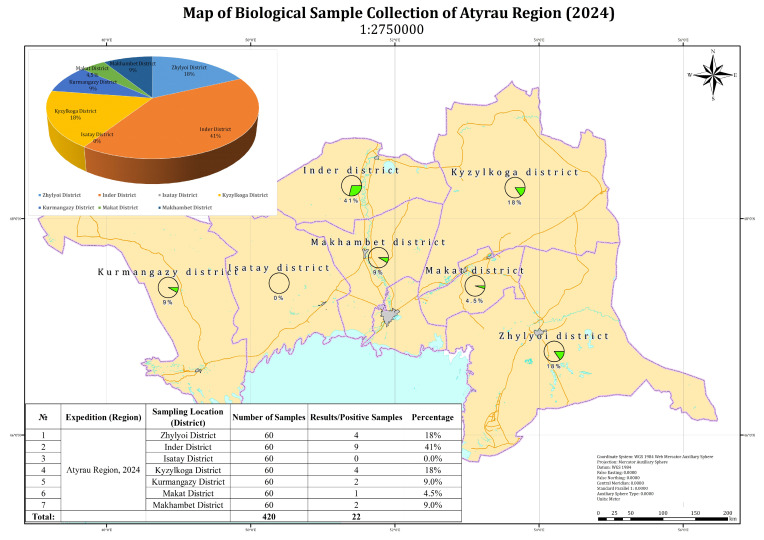
Prevalence of Positive Samples by District in Atyrau Region (2024).

**Figure 3 viruses-17-01229-f003:**
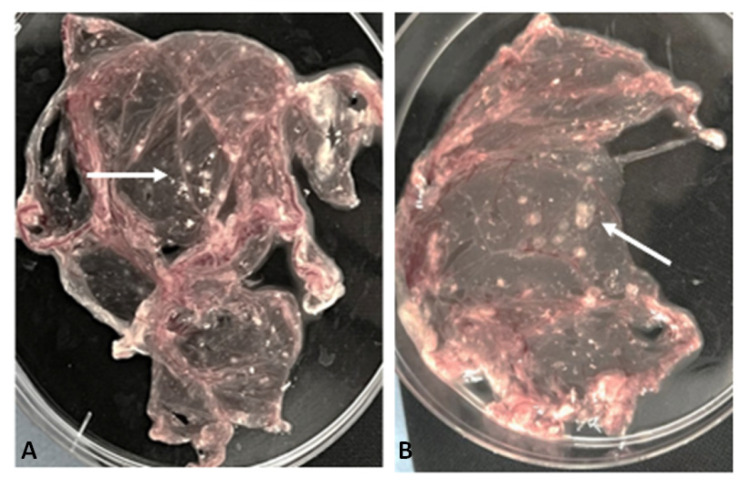
Formation of viral plaques on the CAM of chicken embryos after inoculation with a positive sample. (**A**) Small multiple pockmarks observed after 3 days incubation at 37 °C. (**B**) More pronounced plaques visible after 5 days of inoculation of at 37 °C.

**Figure 4 viruses-17-01229-f004:**
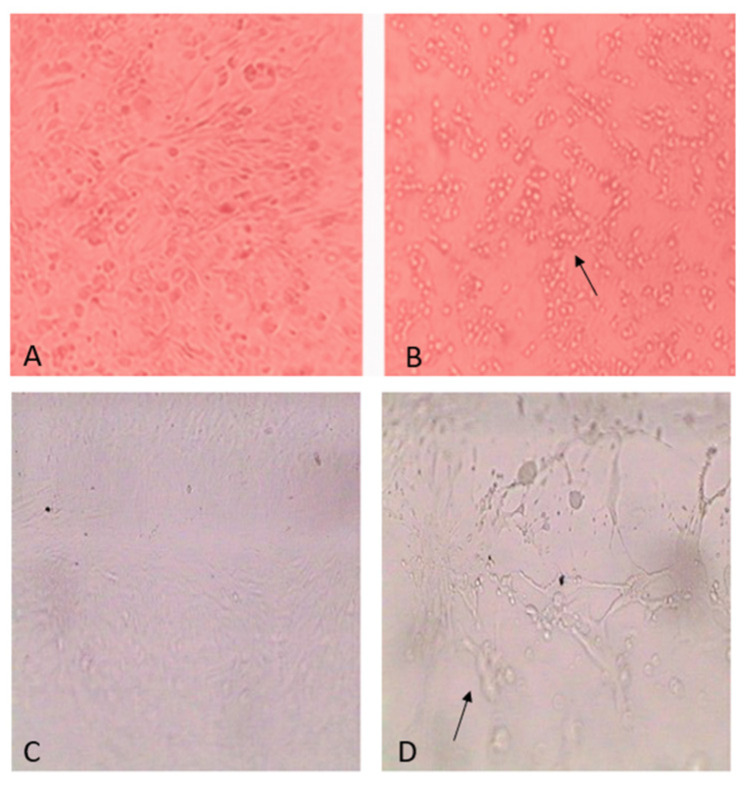
Cytopathic effect of camelpox virus in cell cultures. (**A**) Intact lamb kidney cells; (**B**) Lamb kidney cells exhibiting pronounced CPE 72 h post-inoculation with camelpox virus at 37 °C; (**C**) Intact Vero cells; (**D**) Vero cells displaying characteristic CPE 72 h post-inoculation with camelpox virus at 37 °C (all images at 200× total magnification).

**Figure 5 viruses-17-01229-f005:**
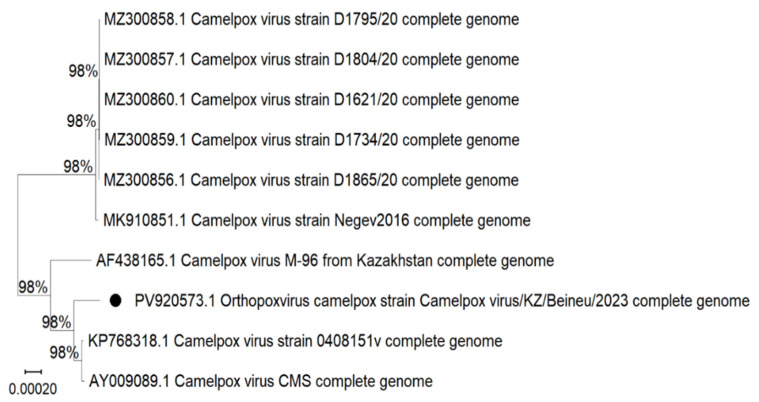
Phylogenetic tree of the *Camelpox virus/Beineu/2023* isolate based on complete genome sequences. The optimal tree with the sum of branch length = 0.003 is shown. The percentage of replicate trees in which the associated taxa clustered together in the bootstrap test (500 replicates) are shown next to the branches. The tree was constructed in MEGA version 12 using the Neighbor-Joining method with the Tamura–Nei model. Evolutionary distances are expressed as the number of substitutions per site, as indicated by the scale bar. The analysis included 10 complete nucleotide sequences (final alignment length: 206,183 positions, with pairwise deletion of ambiguous sites).

**Table 1 viruses-17-01229-t001:** Amino acid substitutions in the A26L, A33R, B5R, and H3L protein sequences compared to other camelpox virus isolates.

Strain	Gene Names								
	H3L		A26L					A33R	B5R
	6	175	236	276	336	374	394	1	308
PV920573.1	R	Y	A	A	L	D	Q	-	N
KP768318.1	R	Y	S	A	V	D	Q	-	N
AY009089.1	R	Y	S	A	V	D	Q	M	N
AF438165.1	R	Y	A	A	V	-	Q	M	N
MK910851.1	K	H	A	S	V	-	H	-	D
MZ300856.1	K	H	A	S	V	-	H	M	D
MZ300859.1	K	H	A	S	V	-	H	M	D
MZ300860.1	K	H	A	S	V	-	H	M	D
MZ300857.1	K	H	A	S	V	-	H	M	D
MZ300858.1	K	H	A	S	V	-	H	M	D

Note: Amino acids are designated by letters in the IUPAC single-symbol system. A—Alanine, D—Aspartic acid, H—Histidine, K—Lysine, L—Leucine, M—Methionine, N—Asparagine, Q—Glutamine, R—Arginine, S—Serine, V—Valine, Y—Tyrosine, “–-—absence of amino acid (gap/deletion).

## Data Availability

The data presented in this study are included in the manuscript or are available on request from the corresponding author. The virus genomes sequenced in this study have been deposited in GenBank using the accession numbers PV920573.1.

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
