# Peer review of "Camelpox Virus in Western Kazakhstan: Isolation and Phylogenetic Analysis of a New Strain"

_viruses, 2025, doi:10.3390/v17091229_

Round 1
Reviewer 1 Report
Comments and Suggestions for Authors
The manuscript by Yerbol Bulatov and colleagues reports the isolation and genomic characterization of a camelpox virus (CMLV) strain obtained from dromedaries in Kazakhstan. The study provides basic virological, molecular, and phylogenetic data on this new isolate and places it in the context of previously characterized CMLV strains. The subject is relevant, as CMLV is an important veterinary pathogen and genomic information remains relatively limited. The manuscript provides a qualitative biological characterization. However, the genomic part of the manuscript does not provide sufficient depth, but can be improved with revision.
The manuscript contains interesting and important data on the geography of the virus, but has a limited sample size for Isolation since the successful virus isolation was achieved from only one sample, despite several PCR-positive samples being identified. This limits the generalizability of the findings regarding viral characteristics and behavior in cell culture. While acknowledged, it’s a notable limitation.
The major concern is related to genomic and phylogenetic analyses. The differences between strains should be discussed - are receptor-binding proteins touched with mutations? The phylogenetic tree (Figure 5) needs to be rooted, it will give the base for the conclusions about the direction of the geographical distribution. Furthermore, the Figure 5 captions should explain the scale and statistics near the nodes. Besides, the authors should explain (or at least comment) the lack of functional studies, since the functional studies (eg, investigating the virulence or immune response to the new isolate) would provide a more comprehensive understanding of the virus.
The minor concern is related to bioinformatic methodology. Software parameters, model selection for phylogeny and software versions must be reported in more detail in the Materials and Methods section.
Overall, this is a well done study but it still needs some improvements.
Author Response
Response to Reviewer 1 Comments
|
||
1. Summary |
|
|
Thank you very much for taking the time to review our manuscript. We sincerely appreciate your comments and constructive feedback. We are grateful for your assistance in improving the quality of our work. Below, you will find our detailed responses to the reviewers’ remarks, as well as the corresponding revisions, which are highlighted in red in the resubmitted files.
|
||
2. Questionsfor General Evaluation |
Reviewer’s Evaluation |
Response and Revisions |
Does the introduction provide sufficient background and include all relevant references? |
Yes |
Thank you for carefully reviewing our manuscript. As there were no comments regarding these sections, no changes have been made in them. |
Are all the cited references relevant to the research? |
Yes |
Thank you for carefully reviewing our manuscript. As there were no comments regarding these sections, no changes have been made in them. |
Is the research design appropriate? |
Yes |
Thank you for carefully reviewing our manuscript. As there were no comments regarding these sections, no changes have been made in them. |
Are the methods adequately described? |
Can be improved |
Thanks for the comment, we agree and in the materials and methods section some minor changes have been made for detail |
Are the results clearly presented? |
Can be improved |
Thanks for the comment, we agree and in the results section some minor changes have been made to improve it. |
Are the conclusions supported by the results? |
Can be improved |
Thank you for the comment. The corresponding additions have been made in the indicated sections. |
3. Point-by-point response to Comments and Suggestions for Authors |
||
Comments 1: The manuscript contains interesting and important data on the geography of the virus, but has a limited sample size for Isolation since the successful virus isolation was achieved from only one sample, despite several PCR-positive samples being identified. This limits the generalizability of the findings regarding viral characteristics and behavior in cell culture. While acknowledged, it’s a notable limitation. |
||
Response 1: Thank you for pointing this out. We agree with this comment. Therefore, we have added sentences in the Discussion section clarifying that this represents a key limitation of our study (11 page, 4 paragraph, 4 line). We acknowledge that if multiple isolates from different sources had been successfully obtained, as initially planned, the study would have been even more valuable and informative. However, during the course of the work, the virus was successfully isolated from only one specimen.
|
||
Comments 2: The major concern is related to genomic and phylogenetic analyses. The differences between strains should be discussed - are receptor-binding proteins touched with mutations? The phylogenetic tree (Figure 5) needs to be rooted, it will give the base for the conclusions about the direction of the geographical distribution. Furthermore, the Figure 5 captions should explain the scale and statistics near the nodes. |
||
Response 2: Thank you for your valuable comments. We fully agree with this comment and have made significant changes to section 3.3. page 7-9. In addition to the whole genome analysis, we analyzed genes encoding proteins involved in virus-host interactions, including A26L, A33R, B5R, and H3L, which are known to play an important role in receptor binding and viral entry in orthomyelopoxviruses (Table 1, page 9). A phylogenetic tree was also constructed for comparison between isolates whose whole genomes are available in the NCBI database. However, we would like to emphasize that conducting an in-depth evolutionary analysis was not initially part of the objectives of our study. The figure legend has been revised in accordance with your recommendations. We carefully considered your comments and came to the conclusion that our topic of the article requires a broader molecular genetic analysis, while the studies performed have certain limitations within the framework of a grant project. In this regard, we propose to adjust the title of the article, making it more general and reflecting the actual volume of work performed: “Camelpox Virus in Western Kazakhstan: Isolation and Phylogenetic Analysis of a New Strain”. We kindly ask you to accept our revisions. Although an in-depth molecular genetic analysis was not performed, we believe that our study has scientific value and contributes to the understanding of camelpox virus circulation. We look forward to your favorable response.
|
||
Comments 3: Besides, the authors should explain (or at least comment) the lack of functional studies, since the functional studies (eg, investigating the virulence or immune response to the new isolate) would provide a more comprehensive understanding of the virus. |
||
Response 3: Thank you for pointing this out. We agree with this observation. Therefore, we have added a clarification in the Discussion section explaining that our study did not conduct functional studies (11 page, 7 paragraphs, 1-4 lines).
|
||
Comments 4: Software parameters, model selection for phylogeny and software versions must be reported in more detail in the Materials and Methods section. |
||
Response 4: Thank you for your comments. We agree with your comment and have made a change to section 2.7 (4 page) which describes Software parameters, model selection for phylogeny in the Materials and Methods section. |
||
4. Response to Comments on the Quality of English Language |
||
The English is fine and does not require any improvement. |
Reviewer 2 Report
Comments and Suggestions for Authors
In the manuscript submitted to me for review entitled "Molecular investigation of camelpox virus isolated from dromedaries in Western Kazakhstan“ the authors present a study identifying potential natural reservoirs of camelpox virus (CMLV) during inter-epizootic periods.
My remarks and recommendations to the authors are:
- In section 2.2. it is not stated where the cell lines used were purchased or provided.
- It is not stated how the cell lines were cultured, for example what culture medium was used and with what supplements.
- Figures 1 and 2 are of poor quality and the captions in them are not readable. Let the quality of these two figures be improved.
- In figure 3 it is not clear what the left and right parts of the figure represent. Let them be labeled A and B and below the figure describe what they represent.
- In the References section, the year of publication should be in bold.
Author Response
Response to Reviewer 2 Comments
|
||
1. Summary |
|
|
Thank you very much for taking the time to review our manuscript. We sincerely appreciate your comments and constructive feedback. We are grateful for your assistance in improving the quality of our work. Below, you will find our detailed responses to the reviewers’ remarks, as well as the corresponding revisions, which are highlighted in red in the resubmitted files.
|
||
2. Questions for General Evaluation |
Reviewer’s Evaluation |
Response and Revisions |
Does the introduction provide sufficient background and include all relevant references? |
Yes |
Thank you for carefully reviewing our manuscript. As there were no comments regarding these sections, no changes have been made in them. |
Is the research design appropriate? |
Yes |
Thank you for carefully reviewing our manuscript. As there were no comments regarding these sections, no changes have been made in them. |
Are the methods adequately described? |
Can be improved |
Thank you for carefully reviewing our manuscript. We agree with your comments and significant changes have been made to the methods sections (2.2 and 2.7). |
Are the results clearly presented? |
Must be improved |
Thanks for the comment, we agree and in the results section some minor changes have been made to improve it. |
Are the conclusions supported by the results? |
Yes |
Thank you for carefully reviewing our manuscript. As there were no comments regarding these sections, no changes have been made in them. |
Are all figures and tables clear and well-presented? |
Must be improved |
Thank you for your comment, we agree and in the results section we have improved the quality of Figures 1 and 2, made changes to the description for the addition in Figure 3. |
3. Point-by-point response to Comments and Suggestions for Authors |
||
Comments 1: In section 2.2. it is not stated where the cell lines used were purchased or provided. It is not stated how the cell lines were cultured, for example what culture medium was used and with what supplements. |
||
Response 1: Thank you for pointing this out. We agree with this comment. We apologize for omitting such important information. Section 2.2 has been updated to include full information on cell cultures (3 page). |
||
Comments 2: Figures 1 and 2 are of poor quality and the captions in them are not readable. Let the quality of these two figures be improved. |
||
Response 2: Thank you for your valuable comments. We fully agree with this comment and we have improved the quality of these two Figures (5, 6 page). |
||
Comments 3: In figure 3 it is not clear what the left and right parts of the figure represent. Let them be labeled A and B and below the figure describe what they represent. |
||
Response 3: Thank you for pointing this out. We agree with this observation. Changes have been made to the description of Figure 3 (7 page). |
||
Comments 4: In the References section, the year of publication should be in bold. |
||
Response 4: Thank you for your comments. According to your recommendations, the year of publication in the literature was highlighted in bold (12 page). |
||
4. Response to Comments on the Quality of English Language |
||
The English is fine and does not require any improvement. |
Round 2
Reviewer 1 Report
Comments and Suggestions for Authors
The revised manuscript has been clearly improved, presenting a more complete, scientifically sound, and well-supported account. The authors have addressed the previous feedback carefully and introduced several important modifications. This study is relevant to specialists in the field and appears suitable for publication.
Reviewer 2 Report
Comments and Suggestions for Authors
The authors have answered all my questions. The corrections I suggested to improve the manuscript have been introduced into the text. Many additions have been made, especially in the Materials and Methods and Results sections, which contribute to a more complete presentation of the research conducted. I have no new questions or remarks for the authors. In my opinion, the results presented in the manuscript will attract the attention of the readers of Viruses.